# (20S)G-Rh2 Inhibits NF-κB Regulated Epithelial-Mesenchymal Transition by Targeting Annexin A2

**DOI:** 10.3390/biom10040528

**Published:** 2020-03-31

**Authors:** Yu-Shi Wang, He Li, Yang Li, Shiyin Zhang, Ying-Hua Jin

**Affiliations:** Key Laboratory for Molecular Enzymology and Engineering of the Ministry of Education, School of Life Sciences, Jilin University, Changchun 130012, China; wangyushi0317@hotmail.com (Y.-S.W.); lihe0607@163.com (H.L.); liyang915@jlu.edu.cn (Y.L.); zsyshiyin@163.com (S.Z.)

**Keywords:** Anxa2, epithelial-mesenchymal transition, NF-κB, (20S)G-Rh2

## Abstract

(1) Background: Epithelial-mesenchymal transition (EMT) is an essential step for cancer metastasis; targeting EMT is an important path for cancer treatment and drug development. NF-κB, an important transcription factor, has been shown to be responsible for cancer metastasis by enhancing the EMT process. Our previous studies showed that (20S)Ginsenoside Rh2 (G-Rh2) inhibits NF-κB activity by targeting Anxa2, but it is still not known whether this targeted inhibition of NF-κB can inhibit the EMT process. (2) Methods: In vivo (20S)G-Rh2-Anxa2 interaction was assessed by cellular thermal shift assay. Protein interaction was determined by immuno-precipitation analysis. NF-κB activity was determined by dual luciferase reporter assay. Gene expression was determined by RT-PCR and immuno-blot. EMT was evaluated by wound healing and Transwell assay and EMT regulating gene expression. (3) Results: Anxa2 interacted with the NF-κB p50 subunit, promoted NF-κB activation, then accelerated mesenchymal-like gene expression and enhanced cell motility; all these cellular processes were inhibited by (20S)G-Rh2. In contrast, these (20S)G-Rh2 effect were completely eliminated by overexpression of Anxa2-K301A, an (20S)G-Rh2-binding-deficient mutant of Anxa2. (4) Conclusion: (20S)G-Rh2 inhibited NF-κB activation and related EMT by targeting Anxa2 in MDA-MB-231 cells.

## 1. Introduction

Playing key roles in embryonic development, epithelial-mesenchymal transition (EMT) facilitates body formation and tissue differentiation [1,2,3,4]. EMT is triggered by a genetic phenotype shift from epithelial-like to mesenchymal-like in response to pleiotropic signal; cells acquire the migratory and invasive properties by modifying adhesion molecules [3,4,5]. During this process, EMT specific transcription factors (EMT-TFs), in company with other regulatory factors like histone modifier and non-coding RNA, modify the gene expression through different states along EMT [6,7,8,9,10,11]. With the transcriptional activation of EMT-TFs, the expression of adherent junction components and tight junction components are negative-regulated, followed by the alteration in cadherin intermediate filament composition and cellular adhesion status [1,3,8,9]. EMT is also implicated in other physiological and pathological processes including would healing, tissue repair, fibrosis and cancer [1,2,3,4,5,6,11,12]. Tumor metastasis is the leading cause for cancer associated mortality; EMT is the main impetus and the essential access within this duration [11,12]. Despite the enhanced migration capability and invasiveness, EMT protects cancer cells from senescence, apoptosis and immuno-response by enhancing stem-cell properties and triggers the metastasis and dissemination to long distance through circulatory systems [13,14]. Focusing on the diverse regulated molecules, multiple drugs have shown their potential as EMT inhibitors by targeting the tumor microenvironment, corresponding extracellular receptors, intracellular kinases and EMT inducing transcription factors [15]. As its well-described clinical signature of EMT in cancer development, therapeutic strategies and targeted drugs towards EMT have been involved in cancer treatment [16,17,18].

NF-κB is an important transcription factor involved in multiple biologic processes as immune response, stress response, apoptosis, cell proliferation and cell metastasis [18]. The abnormal activation of NF-κB has been regarded as a hallmark of cancer, which promotes both tumorigenesis and tumor development [19,20,21]. NF-κB has been shown responsible for the expression of EMT-TFs including SNAIL, TWIST1, SLUG, SIP1 and ZEB1 and promotes EMT progression upon diverse signaling and stimuli, especially in cytokine and chemokine-induced cell growth and migration [22,23]. In the meanwhile, EMT promotes mesenchymal and stem-like phenotype of cancer cells and alters tumor microenvironment via autocrine and paracrine signaling, resulting in a constitutive activation of NF-κB in a ligand-receptor manner rather than genetic alteration [24]. Therefore, inhibition of NF-κB seems to be an important way to inhibit the progression of EMT and related tumors [25,26,27].

In our previous study, (20S)G-Rh2 has been identified as an NF-κB inhibitor targeting Anxa2, a NF-κB p50 subunit binding protein [28]. Here, we investigated the Anxa2 regulation on NF-κB and EMT. We further studied detailed molecular mechanisms for (20S)G-Rh2-induced NF-κB inhibition and the following EMT inhibition in invasive breast cancer cells. The results provide new evidences for the anticancer activity of ginsenosides.

## 2. Materials and Methods

### 2.1. Cell lines and Culture

Human breast cancer cell line MDA-MB-231 (ATCC, HTB-26), MCF-7 (ATCC, HTB-26) and HEK-293-T (ATCC, CRL-11268) were cultured in DMEM high-glucose medium (Gibco) supplemented with 10% fetal bovine serum (Biological Industries, BI), 100 units/mL penicillin and 100 μg/mL streptomycin, in a humidified 5% CO2 atmosphere at 37 °C.

### 2.2. Chemicals, Antibodies and Plasmids

(20S)G-Rh2, etoposide, phorbol 12-myristate 13-acetate (PMA) were purchased from Sigma-Aldrich.

Antibodies for Twist (25465-1-AP), SIP1 (21672-1-AP), Slug (12129-1-AP), Snail 1 (13099-1-AP), MMP-2 (10373-2-AP), MMP-9 (10375-2-AP), Anxa2 (11256-1-AP), β-actin (66009-1-lg) and myc-tag (16286-1-AP, 60003-2-lg) were purchased from Proteintech (Proteintech Group, Inc.). Antibodies for E-cadherin (sc-8426), N-cadherin (sc-59987), NF-κB p50 subunit (sc-7178), Anxa2 (sc-47696) and IgG (sc-2025) were purchased from Santa Cruz Biotechnology. HRP-conjugated goat anti-rabbit IgG (H + L) secondary antibody (31460) and HRP-conjugated goat anti-mouse IgG (H+L) secondary antibody (31430) were purchased from Invitrogen.

Plasmids for over-expression of wild-type Anxa2 (pcs4-Anxa2-WT-myc) and Anxa2 K031A mutant (pcs4-Anxa2-K301A-myc) were shown as described [28]. A truncation with 1–33 deletion of Anxa2 was amplified by polymerase chain reaction (PCR), followed by a recombination into pcs4-myc vector for over-expression of Anxa2-delta N-terminus (pcs4-Anxa2-dN-myc). For lentivirus package, pLVX-TetOne-Puro (Clontech, 631849), pMD2.G (Addgene plasmid # 12259) and psPAX2 (Addgene plasmid # 12260) were gifts from Professor Zhihua Zou (Collage of Life Sciences, Jilin University). C-terminus-Myc-tagged Anxa2 full-length protein as well as K301A mutant and delta-N truncation were amplified from pcs4 vector and recombined into pLVX-TetOne-Pure vector for myc-tagged Anxa2 over-expression with lentivirus. The primers used within were shown in Table 1. Dual luciferase reporter assay were performed with pNF-κB-TA-luc (Beyotime, Shanghai, China) and pRL-CMV (Promega, WI, USA).

### 2.3. Transfection

Lipo3000 (Invitrogen) was used for lentivirus package, transient transfection in MCF-7 cells and dual luciferase reporter assay according to the reagent protocol.

For lentivirus transfection, pLVX, pMD2.G and psPAX2 were co-transfected into HEK-293-T cells and supernatant for 24 h and 48 h was collected then added into MDA-MB-231 cells.

### 2.4. Immuno-Precipitation

50 μL of Protein A/G Magnetic Beads (MCE, K0202) was washed with 400 μL of IP lysis buffer (Pierce) for 3 times. 5 μg of antibody for immune-precipitation was diluted with 500 μL of IP lysis buffer, then added to the prepared beads, followed by a rotation for 2 h at 4 °C. MDA-MB-231 cells and MCF-7 cells were collected and lysed with IP lysis buffer supplemented with Protease Inhibitor Cocktail (Roche, 04693124001) and 1-mM phenylmethanesulfonyl fluoride (PMSF). Cell lysis was centrifuged with 12,000× *g* for 20 min at 4 °C and the supernatant was collected. The antibody bonded beads were then collected and combined with cell lysis containing 500 μg of protein with a final volume of 400 μL, followed by another rotation for 2 h at 4 °C. The beads were then washed with IP lysis buffer for 3 times and collected for immuno-blot analysis.

### 2.5. Cellular Thermal Shift Assay

MDA-MB-231 cells and MCF-7 cells were cultured in 100-mm culture plates until the confluence reached 90%. The culture medium was then replaced with new medium supplemented with 15-μM (20S)G-Rh2 (~10 μg/mL) followed by an incubation for 1 h in a humidified 5% CO_2_ atmosphere at 37 °C. After digested with trypsin (0.25%, *w*/*v* in PBS) and counted, cells were collected with centrifugation at 400× *g* for 5 min and re-suspended with PBS containing 1-mM PMSF to a final cell density of 2 × 10^7^ cells/mL. Each 100 μL of cell suspension was added to a 200-μL tube and heated at indicated temperature for 3 min and incubated at 4 °C for another 2 min. After 2-time rapid freeze-thawing from −80 °C to 25 °C, cell suspension was centrifuged with 20,000× *g* for 20 min at 4 °C, the supernatant was collected for immune-blot analysis.

### 2.6. Dual Luciferase Reporter Assay

pNF-κB-TA-luc and pRL-CMV (10:1, w:w) were co-transfected into MDA-MB-231 cells and MCF-7 cells with Lipo3000 and cultured for 24 h before chemical treatment. The activity of luciferase was determined with Dual-Luciferase® Reporter Assay System (Promega, E1910) according to the manufacture’s protocol. Luminescence generated by luciferase was collected via Infinite F200 Pro (TECAN).

### 2.7. Real-Time Polymerase Chain Reaction

Whole-cell RNA was isolated with TRIzol (Invitrogen). 2 μg of whole-cell RNA was proceeded with High Capacity cDNA Reverse Transcription Kit (Applied Biosystems, 4368814) for cDNA synthesis followed by real-time PCR analysis via PowerUp SYBR Green Master Mix (Applied Biosystems, A25742) and 7500 Real-Time PCR System (Applied Biosystems). The primers used were shown in Table 2. Gene expression was normalized to that of GAPDH and visualized in histogram format.

### 2.8. Cell Migration Analysis

A culture-insert 2 well (Ibidi, 81176) was utilized for would healing assay to determine the migration capability of MDA-MB-231 cells. MDA-MB-231 cells were digested and re-suspended to a final density of 5 × 10^5^ cells/mL. Cell suspension was added into chambers as well as outer area and cultured for 24 h for cell adhesion before the inserts were removed. Then the culture medium was replaced with new medium supplemented with indicated chemicals. Then the wound healing status was recorded by microscopy at 12 h, 24 h and 48 h after culture medium alteration.

### 2.9. Cell Invasion Analysis

A Matrigel Transwell invasion assay was performed to determine the invasiveness of MDA-MB-231 cells. A total of 10^4^ cells in serum-free DMEM medium were seeded into upper chamber (8-μm pore, Corning, 3464) pre-coated with 1 mg/mL Matrigel Matrix (Corning, 354234) and DMEM medium containing 20% FBS was added into 24-well plate. After incubation for 24 h in a humidified 5% CO_2_ atmosphere at 37 °C, cells were fixed with pre-cooled methanol for 30 min and stained with 0.1% crystal violet (w/v) in PBS for 10 min. Images were collected by microscopy to determine the cells passing through Matrigel basement.

### 2.10. Statistical Analysis

All data were obtained from independent triple-replicated experiments and presented as the mean ± standard deviation (SD). Significance was determined by a two-tail Student’s t-test via GraphPad Prism v.6 (GraphPad Inc., USA) and data with *p*-value < 0.05 were considered of statistical significance.

## 3. Results

### 3.1. Anxa2 Bound to NF-κB p50 Subunit in MDA-MB-231 Cells and MCF-7 Cells

It has been well-established that Anxa2 binds to p50 to promote NF-κB activation and cell survival in hepatocellular carcinoma and pancreatic cancer cell lines [28,29]. In order to tell the activity of Anxa2 in NF-κB associated EMT in breast cancer, we first assessed the interaction of Anxa2 with NF-κB p50 subunit in breast cancer MDA-MB-231 and MCF-7 cells by coIP analysis (Figure 1A). As the N-terminus of Anxa2 is responsible for its interaction with p50 [29], we transiently transfected C-terminus myc-tagged full-length Anxa2 and an N-terminus deleted truncated version of Anxa2 (Anxa2-dN). Precipitated by anti-myc antibody, Anxa2-dN failed to interact with p50, demonstrating Anxa2 bound to p50 via its N-terminus in breast cancer cells as does in Hepatoma cells (Figure 1B).

### 3.2. Anxa2 Promoted NF-κB Activation and Associated EMT in Invasive Breast Cancer Cells

We co-transfected cells with Anxa2 and dual luciferase reporter system for NF-κB activity analysis, the transfection of full-length Anxa2 enhanced NF-κB activity whereas that of Anxa2-dN failed (Figure 2A). We then analyzed the NF-κB associated EMT-TFs including SNAIL, SLUG, SIP1 and TWIST1, EMT marker (CDH1, gene encoding E-cadherin; CDH2, gene encoding N-cadherin) and pro-invasion matrix metalloproteinases (MMP2, MMP9). The over-expression of full-length Anxa2 increased the expression of three EMT-TFs such as SLUG, SIP1 and TWIST1 and two MMPs (MMP2, MMP9) in invasive breast cancer MDA-MB-231 cells (Figure 2B, left panel). Up-regulated CDH2 and down-regulated CDH1, presenting the accelerated EMT capability were observed in full-length Anxa2 over-expressing MDA-MB-231 cells (Figure 2B, left panel). Correlated with failed activation of NF-κB, the over-expression of Anxa2-dN caused little expression shift in CDH2 and CDH1 (Figure 2B, left panel). In contrast with MDA-MB-231 cells, less-invasive MCF-7 cells presented no obvious alteration in these migration related gene expression under either full-length Anxa2 or Anxa2-dN truncation over-expression (Figure 2B, right panel). Protein levels of these genes in MDA-MB-231 cells were further determined by immune-blot. The up-regulation of protein levels of E-Cadherin and that of down-regulation of N-cadherin, Twist, slug, SIP1, MMP-2 and MMP-9 were observed in accord with the alteration of mRNA levels (Figure 2C).

Wound healing assay and Transwell invasion assays were then performed with Anxa2-over-expressing MDA-MB-231 cells. Full-length Anxa2 over-expression enhanced the wound healing efficiency and invasiveness through Matrigel basement whereas Anxa2-dN truncation over-expression showed no facilitation (Figure 2D,E).

### 3.3. (20S)G-Rh2 Inhibited NF-κB Activation Targeting Anxa2

(20S)G-Rh2 was proven as a natural small-molecule ligand for Anxa2 and inhibited NF-κB activation by interfering Anxa2-p50 interaction in HepG2 cells [28]. For the purpose of investigating the inhibitory effect toward NF-κB of (20S)G-Rh2 in breast cancer cells, a cellular thermal shift assay was performed with MDA-MB-231 cells and MCF-7 cells; (20S)G-Rh2 increased the thermal stability of Anxa2 in both cell lines (Figure 3A), indicating (20S)G-Rh2 bound to Anxa2 in MDA-MB-231 cells and MCF-7 cells. A following immune-precipitation showed (20S)G-Rh2 inhibited Anxa2-p50 interaction at resting state or under co-treatment with NF-κB activator etoposide or PMA in either MDA-MB-231 cells or MCF-7 cells (Figure 3B). NF-κB activity was determined via a dual luciferase reporter assay; (20S)G-Rh2 inhibited NF-κB activity at resting state and co-treated with etoposide or PMA in both MDA-MB-231 cells and MCF-7 cells (Figure 3C).

### 3.4. (20S)G-Rh2 Inhibited the Migration and Invasion of MDA-MB-231 in a Dose-Dependent Manner

MDA-MB-231 cells and MCF-7 cells were treated with increasing concentration of (20S)G-Rh2 and the result showed that NF-κB activity was down-regulated in a dose-dependent manner upon (20S)G-Rh2 treatment (Figure 4A). A dose-dependent alteration was also appeared in the gene expression analysis of TWIST1, MMP2, MMP9, CDH1 and CDH2, the expression of SLUG and SIP1 presented stable by (20S)G-Rh2 in MDA-MB-231 cells; only Snail1 presented an unexpected increase (Figure 4B, left panel). No regulated shift was seen in MCF-7 cells under (20S)G-Rh2 treatment (Figure 4B right panel). The protein levels of EMT-TFs, MMPs and N-cadherin were down-regulated and that of E-cadherin was up-regulated in a (20S)G-Rh2-dose-dependent manner in MDA-MB-231 cells (Figure 4C). Wound healing assay and Transwell invasion assay were performed MDA-MB-231 cells upon various concentration of (20S)G-Rh2 treatment. (20S)G-Rh2 inhibited the migration and invasion of MDA-MB-231 cells in a dose-dependent manner (Figure 4D,E).

### 3.5. (20S)G-Rh2 Binding-Deficient Mutant of Anxa2 Protected MDA-MB-231 Cells from (20S)G-Rh2 Induced NF-κB Inhibition

To provide further evidence for the (20S)G-Rh2 effect on Anxa2 function, we over-expressed the (20S)G-Rh2-binding-deficient mutant, Anxa2-K301A, in MDA-MB-231 cells. A cellular thermal shift assay showed that Anxa2-K301A failed to interact with (20S)G-Rh2 (Figure 5A). (20S)G-Rh2 failed to interfere the interaction between Anxa2-K301A and p50; it even enhanced it (Figure 5B). (20S)G-Rh2 induced NF-κB inhibition and downstream gene expression shift were also unchanged or reversed in Anxa2-K301A over-expressing cells (Figure 5C,D). Cell migration capability and invasiveness were unaltered with Anxa2-K301A overexpression under (20S)G-Rh2 treatment (Figure 5E,F).

## 4. Discussion

Cancer metastasis present responsibility for over 90% cancer mortality. EMT—one of the main generators for cancer metastasis—has become an assignable factor in both cancer research and treatment [1,11]. Derived by EMT promoting genes, epithelial-like cancer cells re-program gene expression phenotype, lose epithelial feature, dissociate from primary tumor tissue and spread with circulatory system [2,4,5,6]. A variety of gene transcriptional regulation was involved in this program, among which EMT-TFs and up-stream transcriptional factors like NF-κB and STAT3, directly alter the expression of key effector molecules [22,23,30,31].

Among them, multi-functional protein Anxa2 behaves as an implicated promoter for EMT and related cellular event. Anxa2 was first associated with cell migration for its heterotetramer with S100A10 [32,33], which promotes extracellular matrix degradation by activating plasmin and following researches indicate Anxa2 is responsible for membrane dynamics in invasive cells by binding to other S100 protein like S100A4 [34], S100A6 [35] and S100A11 [36,37]. Implicated in signaling transduction, phosphorylated Anxa2 at Tyr23 binds to STAT3 and promotes its phosphorylation at Tyr705 as well as following dimerization and nuclear translocation, resulting in up-regulated MMPs and enhanced EMT [38,39]. In addition, couple of researches have demonstrated Anxa2 content is correlated to cell migration capability [40,41,42,43].

It has been well-established that NF-κB bound to the promoter region of ZEB-1/2, SNAIL, SLUG, SIP1 and TWIST1 and the transcription of ZEB-1/2, SLUG, SIP1 and TWIST1 accords with NF-κB activation [22,23]. Anxa2, acting as an NF-κB co-activator in hepatocellular carcinoma and pancreatic cancer [28,29], was first identified bound to p50 subunit of NF-κB in breast cancer cells (Figure 1A,B). Over-expression of Anxa2 in invasive breast cancer MDA-MB-231 cells accelerated NF-κB activation (Figure 2A) and the expression EMT-TFs including SLUG, SIP1 and TWIST1 (Figure 2B,C), followed by the down-regulation of E-cadherin and up-regulation of N-cadherin (Figure 2B,C). Though containing NF-κB binding site, SNAIL somehow failed to follow NF-κB regulation, presenting differential variation trend as other EMT-TFs (Figure 2B and Figure 4B) [23]. MMPs responsible for the degradation of extracellular matrix and promote cell motility appeared to be up-regulated in according with EMT phenotype in Anxa2-over-expressing cells (Figure 2B,C). These gene expression alteration resulted in the enhanced migratory properties and invasiveness (Figure 2D,E). In contrast, no altered expression of EMT-TFs was observed in company with the activation of NF-κB in less aggressive MCF-7 cells (Figure 2A,B), indicating EMT-TFs expression requires additional regulatory mechanism, which is necessary to perform further investigation in the future study.

(20S)G-Rh2 is one of the most famous ginsenosides from ginseng extract for its remarkable anti-cancer activity. (20S)G-Rh2 has been proven efficiently inducing growth arrest and apoptosis [44,45,46,47]. Moreover, in-vivo study indicates that (20S)G-Rh2 inhibits tumor growth by inhibiting tumor associated angiogenesis and enhancing anti-tumor immunological response [48,49,50]. In addition, couples of studies have indicated (20S)G-Rh2 presents inhibitory effect towards EMT. Based on our previous research on (20S)G-Rh2-Anxa2 interaction, we supposed that (20S)G-Rh2 inhibited EMT by targeting Anxa2. A cellular thermal shift assay verified the intracellular interaction between Anxa2 and (20S)G-Rh2 (Figure 3A): an immuno-precipitation as well as NF-κB luciferase reporter assay confirmed the inhibitory impact on NF-κB by (20S)G-Rh2 (Figure 3B,C and Figure 4A) in breast cancer cell line MDA-MB-231 and MCF-7. As a result, down-stream pro-EMT genes were down-regulated with epithelial phenotype marker E-cadherin accelerated in MDA-MB-231 cells under (20S)G-Rh2 treatment (Figure 4B,C); cell migration and invasion were also apparently inhibited (Figure 4D,E).

In our previous work, we have identified the Lys301 residue responsible for (20S)G-Rh2 binding; Anxa2-K301A has been shown to be a (20S)G-Rh2 binding deficiency mutant [28]. In addition, Anxa2-K301A mutants maintained p50 binding capacity and promoted NF-κB under (20S)G-Rh2 treatment [28]. Anxa2a-WT and K301A mutants were transfected with MDA-MB-231 cells to examine the targeted inhibition of NF-κB and related EMTs. (20S)G-Rh2 failed to interact with Anxa2-K301A (Figure 5A); p50-Anxa2-K301A interaction and NF-κB activation were surprisingly enhanced under (20S)G-Rh2 treatment (Figure 5B,C). Functioning as a multi-targeted chemical, (20S)G-Rh2 induced various cellular stress and NF-κB stayed inactive under the targeted inhibition [28,44,45,46,47,48,49,50]. This status was reversed when Anxa2 mutant maintained the activation property regardless of (20S)G-Rh2 and stress related NF-κB activation was generated. Under the constitutive NF-κB activation by Anxa2-K301A over-expression presented higher mesenchymal-like phenotype (Figure 5D) and equivalent migratory and invasive properties (Figure 5E,F).

## 5. Conclusions

Our data suggest that Anxa2 is an important co-activator of NF-κB in breast cancer cells. Over-expression of Anxa2 promoted NF-κB activation and pro-EMT gene expression. With the up-regulation of EMT-TFs and MMPs, MDA-MB-231 cells presented a mesenchymal-like phenotype and achieved enhanced migratory properties as well as invasiveness. (20S)G-Rh2 induced NF-κB inhibition by targeting Anxa2 and led to the suppression in pro-EMT gene expression, resulting in the EMT arrest at the epithelial state. Over expression of Anxa2-K301A maintained NF-κB activation and the mesenchymal-like phenotype under (20S)G-Rh2 treatment in MDA-MB-231 cells. Taken together, (20S)G-Rh2 targets Anxa2 in invasive breast cancer cells, inhibits NF-κB activity and EMT and may be a potent candidate for breast cancer treatment and related research.

## Figures and Tables

**Figure 1 biomolecules-10-00528-f001:**
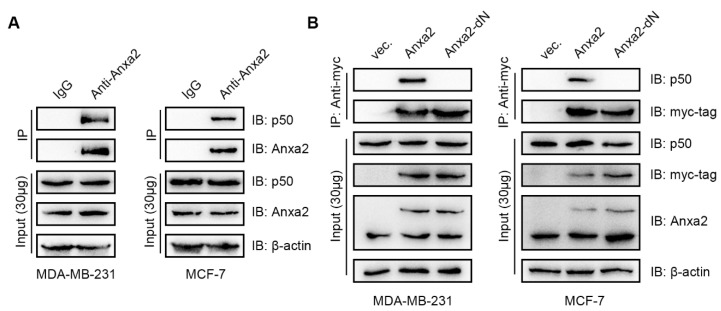
Anxa2 interacts with p50 in breast cancer cell lines. (**A**) An immuno-precipitation was performed with anti-Anxa2 antibody in protein extract from MDA-MB-231 cells and MCF-7 cells and immuno-precipitation with IgG was used as a negative control. (**B**) An immuno-precipitation was performed with anti-myc-tag antibody in protein extract from MDA-MB-231 cells and MCF-7 cells transfected with empty vector (vec.), Anxa2 and Anxa2-dN. A total of 30 μg of whole-cell protein extract was loaded as input. All experiments were repeated for three times.

**Figure 2 biomolecules-10-00528-f002:**
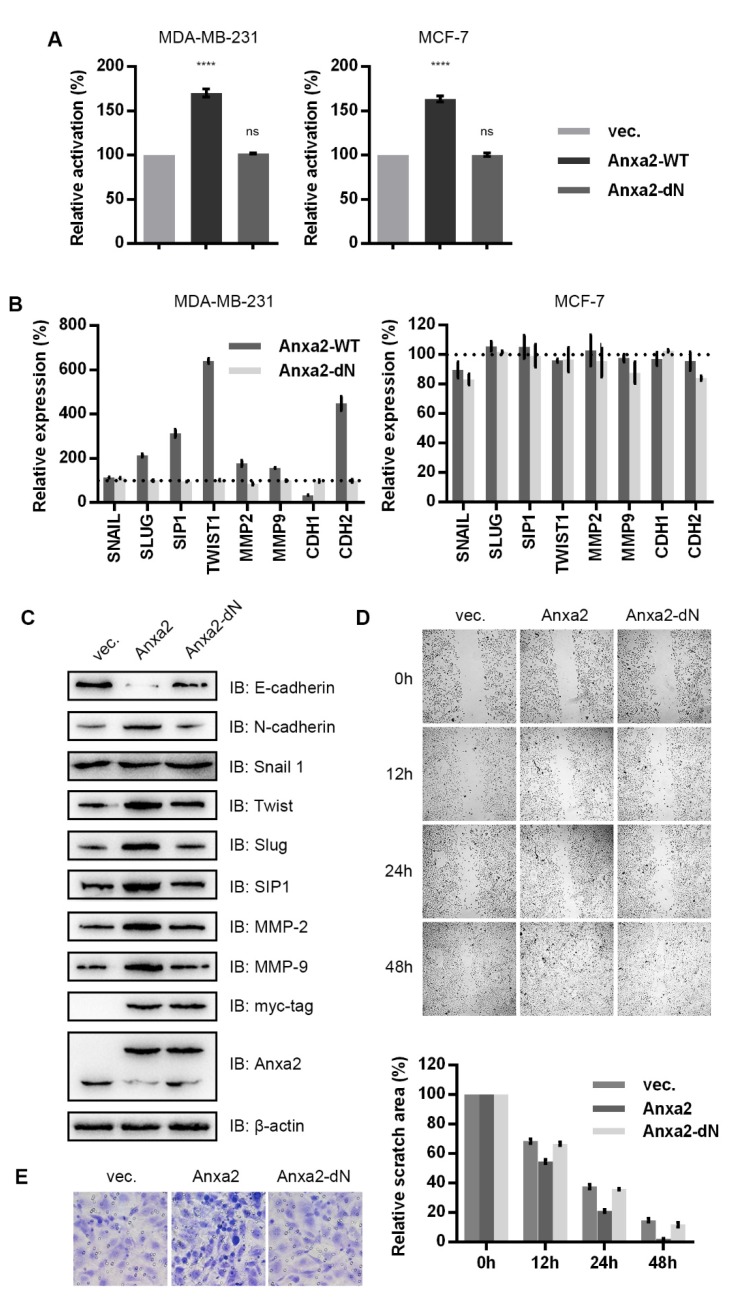
Anxa2 over-expression enhanced NF-κB activation and related cell migration and invasion. (**A**) NF-κB activity was determined via a dual luciferase reporter assay in MDA-MB-231 cells and MCF-7 cells transfected with empty vector, Anxa2 and Anxa2-dN. (**B**) mRNA levels of indicated genes were determined by RT-PCR with RNA extract from MDA-MB-231 cells and MCF-7 cells transfected with Anxa2 and Anxa2-dN t, the dot line at 100% presents mRNA levels in cells transfected with empty vector. (**C**) The protein levels of indicated genes were determined by immuno-blot in MDA-MB-231 cells transfected with empty vector (vec.), Anxa2 and Anxa2-dN. A total of 30 μg of whole-cell protein extract was loaded for immuno-blotting assay. (**D**) A wound healing assay was performed with MDA-MB-231 cells transfected with empty vector (vec.), Anxa2 and Anxa2-dN. Relative scratch area was shown below the microscopy images. (**E**) A Transwell invasion assay was performed with MDA-MB-231 cells transfected with empty vector (vec.), Anxa2 and Anxa2-dN. All experiments were repeated for three times and all data are shown as mean ± SD, with **** presenting *p* < 0.0001, *** presenting *p* < 0.001, ** *p* < 0.01, * presenting *p* < 0.05 and ns presenting *p* > 0.05.

**Figure 3 biomolecules-10-00528-f003:**
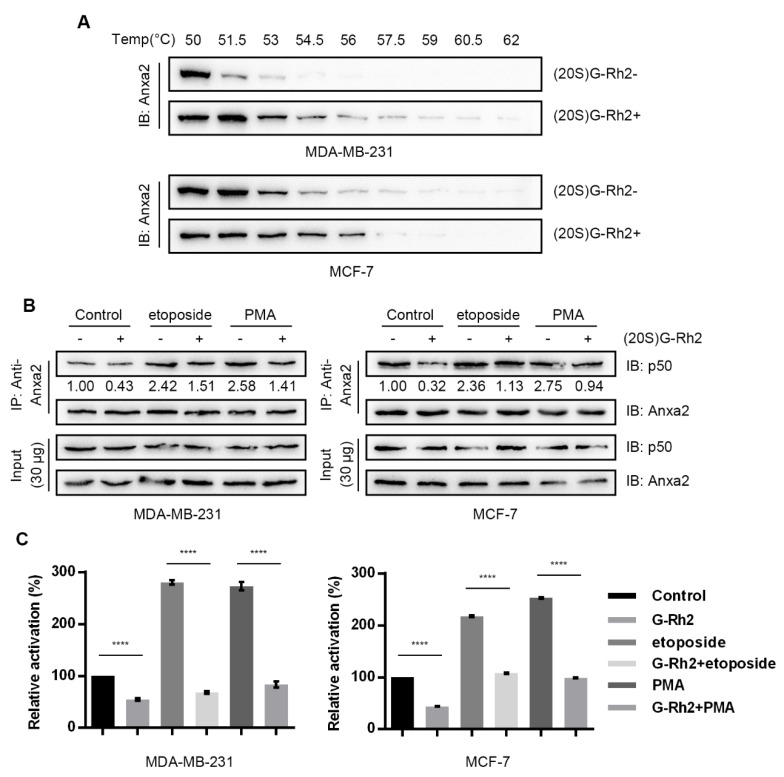
(20S)G-Rh2 inhibits NF-κB activation by binding to Anxa2. (**A**) A cellular thermal shift was performed in MDA-MB-231 cells and MCF-7 cells under 10-μM-(20S)G-Rh2 treatment or not. (**B**) An immuno-precipitation was performed with anti-Anxa2 antibody in protein extract from MDA-MB-231 cells and MCF-7 cells under treatment or 6-μM (20S)G-Rh2, 25 μg/mL etoposide,100 ng/mL PMA and combined chemicals for 12 h. (**C**) NF-κB activity was determined via a dual luciferase reporter assay in MDA-MB-231 cells and MCF-7 cells under treatment or 6-μM (20S)G-Rh2, 25 μg/mL etoposide,100 ng/mL PMA and combined chemicals for 12 h. A total of 30 μg of whole-cell protein extract was loaded as input. All experiments were repeated for three times and all data are shown as mean ± SD, with **** presenting *p* < 0.0001, *** presenting *p* < 0.001, ** *p* < 0.01, * presenting *p* < 0.05 and ns presenting *p* > 0.05.

**Figure 4 biomolecules-10-00528-f004:**
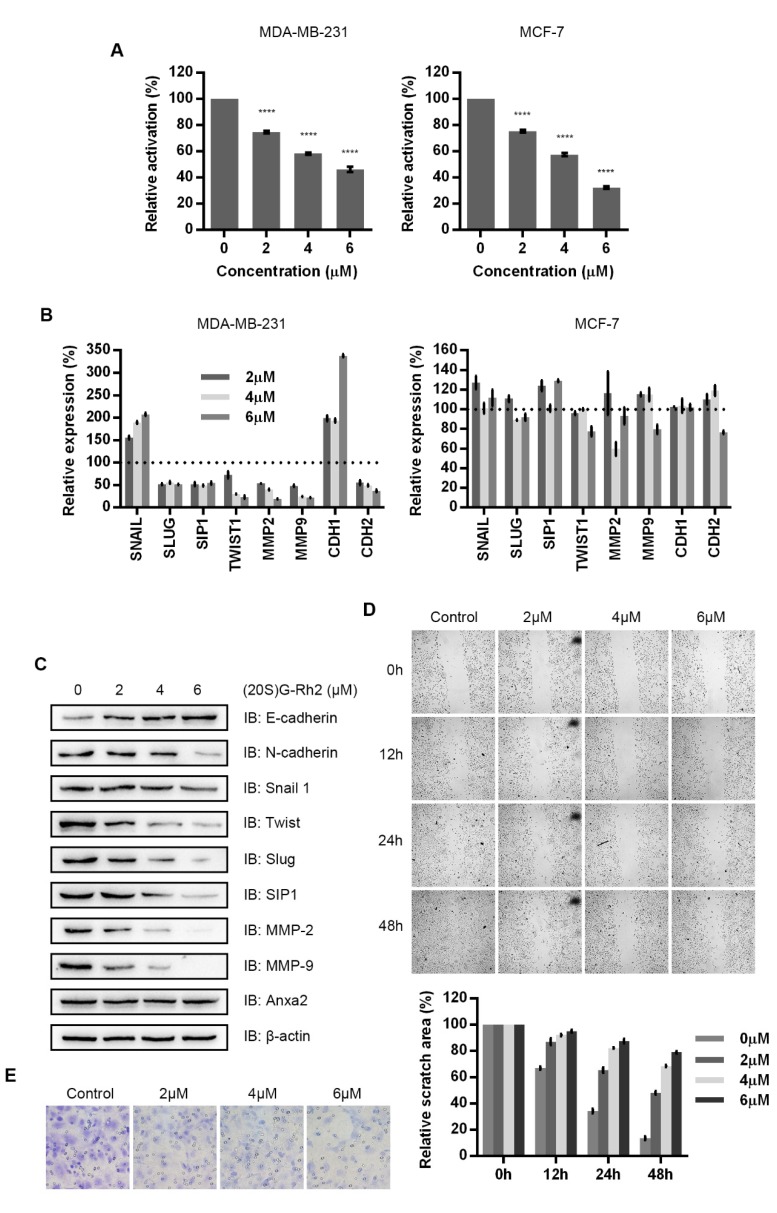
(20S)G-Rh2 inhibits NF-κB related cell migration and invasion. (**A**) NF-κB activity was determined via a dual luciferase reporter assay in MDA-MB-231 cells and MCF-7 cells under (20S)G-Rh2 treatment for 12 h. (**B**) mRNA levels of indicated genes were determined by RT-PCR with RNA extract from MDA-MB-231 cells under (20S)G-Rh2 treatment for 12 h. (**C**) The protein levels of indicated genes were determined by immuno-blot in MDA-MB-231 cells under (20S)G-Rh2 treatment for 12 h. A total of 30 μg of whole-cell protein extract was loaded for immuno-blotting assay. (**D**) A wound healing assay was performed with MDA-MB-231 cells under (20S)G-Rh2 treatment for 12 h. Relative scratch area was shown below the microscopy images. (E) A Transwell invasion assay was performed with MDA-MB-231 cells t under (20S)G-Rh2 treatment for 12 h. All experiments were repeated for three times and all data are shown as mean ± SD, with **** presenting *p* < 0.0001, *** presenting *p* < 0.001, ** *p* < 0.01, * presenting *p* < 0.05 and ns presenting *p* > 0.05.

**Figure 5 biomolecules-10-00528-f005:**
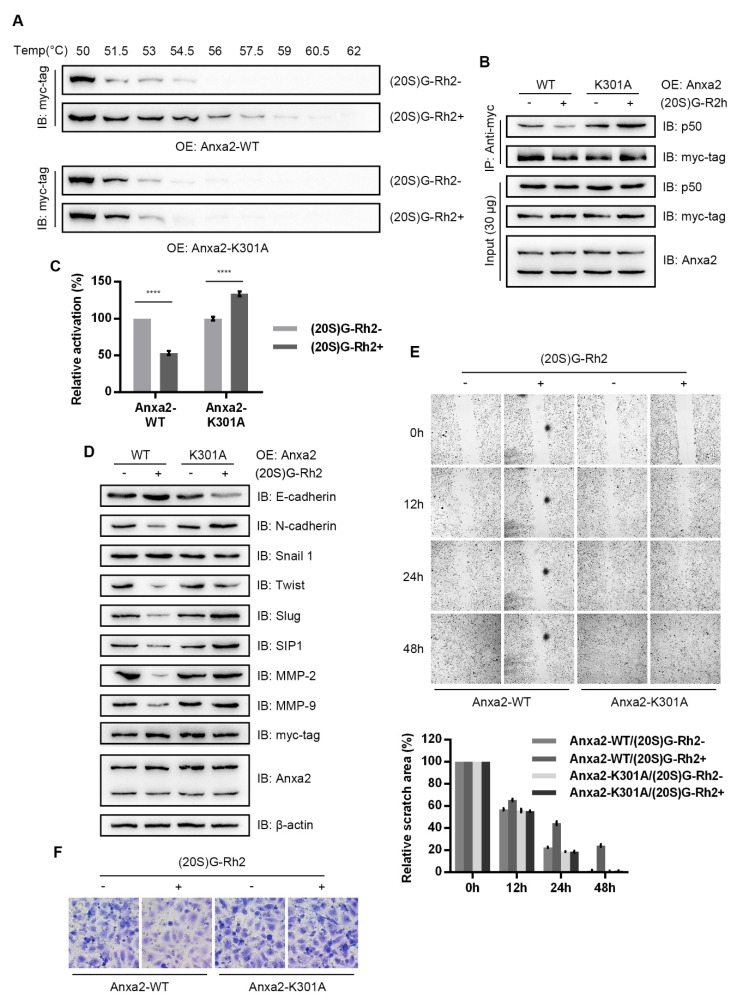
Anxa2-K301A protects MDA-MB-231 cells from (20S)G-Rh2-induced NF-κB inhibition. (**A**) A cellular thermal shift was performed in MDA-MB-231 cells transfected with Anxa2-WT and Anxa2-K301A under 10-μM-(20S)G-Rh2 treatment or not. (**B**) An immuno-precipitation was performed with anti-myc-tag antibody in protein extract from MDA-MB-231 cells transfected with Anxa2-WT and Anxa2-K301A under 6-μM-(20S)G-Rh2 treatment or not. A total of 30 μg of whole-cell protein extract was loaded as input. (**C**) NF-κB activity was determined via a dual luciferase reporter assay in MDA-MB-231 cells transfected with Anxa2-WT and Anxa2-K301A under 6-μM-(20S)G-Rh2 treatment for 12 h or not. (**D**) The protein levels of indicated genes were determined by immuno-blot in MDA-MB-231 cells transfected with Anxa2-WT and Anxa2-K301A under 6-μM-(20S)G-Rh2 treatment or not. A total of 30 μg of whole-cell protein extract was loaded for immuno-blotting assay. (**E**) A wound healing assay was performed with MDA-MB-231 cells transfected with Anxa2-WT and Anxa2-K301A under 6-μM-(20S)G-Rh2 treatment or not. Relative scratch area was shown below the microscopy images. (**F**) A Transwell invasion assay was performed with MDA-MB-231 cells transfected with Anxa2-WT and Anxa2-K301A under 6-μM-(20S)G-Rh2 treatment or not. All experiments were repeated for three times and all data are shown as mean ± SD, with **** presenting *p* < 0.0001, *** presenting *p* < 0.001, ** *p* < 0.01, * presenting *p* < 0.05 and ns presenting *p* > 0.05.

**Table 1 biomolecules-10-00528-t001:** Primers for vector construction.

Primer Description	Sequence
pcs4-Anxa2-dN-F	5’- TTGGATCCATGGATGCTGAGCGGGATGCTTTG -3’
pcs4-Anxa2-dN-R	5’- CCGCTCGAGTCATCTCCACCACACAGGTACAG -3’
pLVX-Anxa2-dN-F	5’- GCGTATACATGGATGCTGAGCGGGATGCTTTG -3’
pLVX-Anxa2-F	5’- GCGTATACATGTCTACTGTTCACGAAATCCT -3’
pLVX-Anxa2-R	5’- GCGGATCCTAGCTATCTAGAGGCTCGAGAGG -3’

**Table 2 biomolecules-10-00528-t002:** Primers for RT-PCR.

Gene Name	Sequence
SNAIL-F	5’- TGCCCTCAAGATGCACATCCGA -3’
SNAIL-R	5’- GGGACAGGAGAAGGGCTTCTC -3’
SLUG-F	5’- ATCTGCGGCAAGGCGTTTTCCA -3’
SLUG-R	5’- GAGCCCTCAGATTTGACCTGTC -3’
SIP1-F	5’- GAGTTGATGCCTCGGCTATTGC -3’
SIP1-R	5’- CTGGACATTGAGCTGCTTCGATC -3’
TWIST1-F	5’- GCCAGGTACATCGACTTCCTCT -3’
TWIST1-R	5’- TCCATCCTCCAGACCGAGAAGG -3’
MMP2-F	5’- AGCGAGTGGATGCCGCCTTTAA -3’
MMP2-R	5’- CATTCCAGGCATCTGCGATGAG -3’
MMP9-F	5’- GCCACTACTGTGCCTTTGAGTC -3’
MMP9-R	5’- CCCTCAGAGAATCGCCAGTACT -3’
CDH1-F	5’- GCCTCCTGAAAAGAGAGTGGAAG -3’
CDH1-R	5’- TGGCAGTGTCTCTCCAAATCCG -3’
CDH2-F	5’- CCTCCAGAGTTTACTGCCATGAC -3’
CDH2-R	5’- GTAGGATCTCCGCCACTGATTC -3’
GAPDH-F	5’- GTCTCCTCTGACTTCAACAGCG -3’
GAPDH-R	5’- ACCACCCTGTTGCTGTAGCCAA -3’

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
