# Peer review of "(20S)G-Rh2 Inhibits NF-κB Regulated Epithelial-Mesenchymal Transition by Targeting Annexin A2"

_biomolecules, 2020, doi:10.3390/biom10040528_

Round 1

Reviewer 1 Report

The authors explore in this manuscript the relationship between G-Rh2 interaction with AnxA2, inhibition of the NFkB pathway and the epithelial-mesenchymal transition. This manuscript is a good follow-up article to the one published in Scientific Reports by the same authors where they describe the relation of this G-Rh2/ANXA2 interaction in cellular death pathway. The description of how ANXA2 associates with G-Rh2 and interfere with NFkB is well done, but for those that previously read the Scientific Reports article it loses a bit of novelty. This does not make the manuscript less relevant.

Material and Methods section is sufficiently detailed and well writen with only one exception. Line 128 is not confusing. Should be better explained. Line 149 data is misspelled as date.

Results are clear and well presented. However there are two issues that can be solved:

1- The authors do not mention how many replicates were done for each experiment in the figure legends. This must be stated for each experiment in every Figure Legend. 

2- Wound healing and invasion assays were not quantified. The authors not only do not mentioned how many times these experiments were replicated, there is no quantification of the results. Invasion assays can be inferred visually by color intensity. But wound healing is not possible to be presented without any quantification. The results highlighted by the authors is not completely clear and hard to assess from the phase contrast images presented. A graph with the percentage of wound closure through time considering the gap at 0h as 100% is needed.

I liked the rationale of the manuscript and how the results fit in this story.

In Figure 4B the is an expressive dose dependent increase in the amount of SNAIL and CDH1. This is rather unexpected. Authors did not describe this in their results and did not provide any possible explanation for it. It would be very good to add this information in the description of the results and also to discuss about it.

Lines 173 and 176 CDH is misspelled as CHD

There are too many confusing sentences in the introduction and discussion. I would recommend a new round of English language and style editing to the article. These sentences make the reader to lose his/her focus and makes the manuscript lose in appeal. I will highlight some examples:

Line 33: "... modify the plasticity of gene expression" . Plasticity is not the ideal word for this sentence. I would take it out and leave only "modify gene expression..."

Lines 35-36: "followed by the alteration in cadherin intermediate filament expression" This sentence is confusing. What changes? Cadherin expression or intermediate filaments?

Line 42: "Focusing on the individual pathway,..." No individual pathway was mentioned before in the text so the start of this sentence makes no sense.

Line 45-46: The sentence is very confusing, I could not understand what the authors wanted to say.

Line49-50: "... promotes tumorigenesis, development and progression of human cancer" This seems a bit redundant considering the meaning of tumorigenesis. I recommend rephrasing it.

Line 53 starts with "On the other hand,..."this is used to show a contrasting idea with what was previously presented. This is not the case here. So the phrase should start with a different expression.

The first sentence in the discussion is not clear. What the authors meant by "EMT facilitates as the main generator for cancer metastasis..." Besides being confusing this is not entirely true as we already know that EMT might not be necessary for metastatic colonisation.

It would be very interesting if the authors could provide a hypothesis for why G-Rh2 treatment enhanced the effects of NFkB activation.

But the main issue of the discussion is the absence of a paragraph to highlighted what has already been shown regarding ANXA2 and EMT. There are several works from Zheng L. in pancreatic cancer, Rocha MR. in colon cancer and one of the most relevant for the context of this manuscript is the work from Wang T (doi: 10.18632/oncotarget.5199) that explores the role of ANXA2 activating STAT3 and leading to EMT in breast cancer cells. To explore ANXA2 and EMT in the discussion will highlight the importance of the findings in this manuscript.

Conclusion is well supported by the results obtained. I would rephrase the sentence in lines 314-315. It is somewhat confusing. What is a comprehensive suppression? Sometimes qualificatives are unnecessary and bring more confusion than clarification.

The abstract has similar mistakes to the ones that were already highlighted here. After some English language style revision it will be great.

In overall I find this manuscript, novel, well constructed and worthy of publication after some minor changes mentioned above and especially after English language and style editing.

Author Response

Dear reviewer:

Thanks for your kind comments and recommendation for the paper, and we will take all your advice in the revised version of the manuscript. Detailed modification is listed below.

Comment: Miss-spelling in Material and Methods.

Answer: Miss-spelling in line 128 and line 149 has been corrected.

Comment: Experiment replication.

Answer: Notation for experiment replication has been added within all figure legends.

Comment: Would healing assay quantification.

Answer: The scratch area has been measured and a histogram has been added below each would healing image.

Comment: Miss-spelling of CDH1

Answer: All miss-spelling has been corrected.

Comment: About SNAIL

Answer: It is reported SNAIL contains NF-kappa B binding sites but could not follow NF-kappa B regulation (Ref 23, doi: 10.1371/journal.pone.0169622.), which matches our results. There should be more regulation not accessible for now beyond RNA and protein level. The description and explanation has been added in the paper.

Comment: Confusing sentences

Answer: All confusing sentences mentioned has been arranged following the recommendation.

Comment: About Anxa2 and EMT

Answer: An additional paragraph has been inserted with discussion section to describe the relationship between Anxa2 and EMT.

Comment: Confusing sentences in conclusion

Answer: All confusing sentences mentioned has been arranged following the recommendation.

Reviewer 2 Report

This manuscript deals with the role of Annexin A2 (AnxA2) in epithelial-mesenchymal transition (EMT) using two human breast cancer cells, MCF-7 and the more aggressive metastatic MDA-MB-231 cell line. A compound isolated from ginseng is described to bind to AnxA2 and inhibit its interaction with the NF-kB p50 subunit, which is important for promoting cell migration and EMT. Thus, the (20S)G-Rh2 compound from ginseng may be a future drug candidate to treat aggressive triple-negative breast cancers. Therefore, this manuscript merits publication.

Comments

Annexin A2 should not be abbreviated in the title

I will comment Figure by Figure

Figure 1

I am very surprised that that level of AnxA2 in MCF-7 appears to be equal or higher than in MDA-MB-231 cells, which is not found by others (f ex  doi: 10.1371/journal.pone.0044299; a paper that should have been cited). The level of AnxA2 in MCF-7 is very low. One of the reasons why MDA-MB-231 are so metastatic may be the high levels of AnxA2; which has been implicated in metastasis and angiogenesis – two processes related to cell mobility. Input; how many µg loaded should be indicated (this will also allow for comparisons between the two cell lines), and also selected Mr of standard proteins. Loading controls should be added.

When AnxA2 is truncated by 33 aa, one would expect a decrease in Mr of about 4 kDa and this can be observed by SDS-PAGE and Western blotting for wild type AnxA2. This is not seen in Panel B; what is the Mr of AnxA2 with a Myc-tag? Could it be that this is not seen due to the larger Mr?

Figure 2

Only a small comment; transfection with vector alone (without insert) seems to increase the level of endogenous AnxA2 (often seen), not seen in Figure 1. Again, indicate how many µg were loaded.

Figure 3

For Panel A and input, Panel B, it should be indicated how many µg were loaded.

To me, the data in Panel A would be stronger if a shift in Tm can be seen by circular dichroism of purified AnxA2 +/- the compound. However, I do not ask for this in this manuscript, but it would make a stronger case. In addition, isothermal titration calorimetry experiments were carried out previously (ref 28 in this manuscript).

Figure 4

For Panel C, it should be indicated how many µg were loaded.

Figure 5

AnxA2-K301A is AnxA2-K302 in your ref 28; I guess the latter indicates the presence of Met and the first that the sequence starts with Ser (acetylated). It is nowhere mentioned that the K is also part of the mRNA-binding site of AnxA2 in domain IV of the core structure (ref Aukrust et al., 2007; DOI: 10.1016/j.jmb.2007.02.094).

For Panels A, B (input) and D, it should be indicated how many µg were loaded.

This manuscript deals with the role of Annexin A2 (AnxA2) in epithelial-mesenchymal transition (EMT) using two human breast cancer cells, MCF-7 and the more aggressive metastatic MDA-MB-231 cell line. A compound isolated from ginseng is described to bind to AnxA2 and inhibit its interaction with the NF-kB p50 subunit, which is important for promoting cell migration and EMT. Thus, the (20S)G-Rh2 compound from ginseng may be a future drug candidate to treat aggressive triple-negative breast cancers. Therefore, this manuscript merits publication.

Author Response

Dear reviewer:

Thanks for your kind comments and recommendation for the paper, and we will take all your advice in the revised version of the manuscript. Detailed modification is listed below.

Comment: Annexin A2 should not be abbreviated in the title.

Answer: Full name of Annexin A2 has been used in the title.

Comment: Figure 1.

Answer:

  1. Loading quantity has been added in both figure and legend.
  2. About the protein level of Anxa2 in MDA-MB-231 cell line and MCF-7 cell line.

The immuno-blotting for MDA-MB-231 and MCF-7 were not prepared simultaneously or performed for compression, and the blot itself could not present protein content in different cell lines. The selectivity for target genes of NF-kappa B should the determining factor for the well-described differential metastatic capability, and we believe Anxa2-p50 interaction only regulate the activity, probably nuclear entering, but not target selection. So no experiments are arranged for MDA-MB-231 and MCF-7 compression.

  1. About the molecular weight for Anxa2 and Anxa2-dN.

There should be an obvious shift between Anxa2 and Anxa2-dN. However, only tiny difference is observed in our experiment. This may be caused by multiple modifications at the N-terminus of Anxa2, especially at Ser1, Cys7, Tyr24 and Ser26, which may present a shifted apparent molecular weight. We believe it is a coincidental apparent molecular weight in immuno-blotting assay.

Comment: Figure 2

Answer:

  1. Loading quantity has been added in figure legend.
  2. The transfection as well as the over-expression is not that sufficient in figure 1, and a sufficient over-expression may inhibit the endogenous expression of Anxa2.

Comment: Figure 3

Answer:

  1. Loading quantity has been added in both figure and legend.
  2. Cellular thermal shift assay is performed with identical cell amount and all samples are prepared simultaneously and strictly parallelly. As thermal denaturation affects the solubility of proteins, only parts of proteins stay soluble and no exact loading quantity can be provided for cellular thermal shift assay.
  3. Thanks for your recommendation again. As ITC and thermal shift assay present similar results for Anxa2-(20S)G-Rh2 interaction, it has been basically identified Anxa2 is the cellular target of (20S)G-Rh2. So no more description for this interaction is not arranged and provided.

Comment: Figure 4

Answer: Loading quantity has been added in figure legend.

Comment: Figure 5

Answer:

  1. About K301. It is indeed that Anxa2 sometimes is counted from the second amino acid as Ser1 for the removal of original Met1. In the paper, no prokaryotic system is utilized and no Met1 is needed, and we choose this form for presentation.
  1. Though capable for DNA and RNA binding, there is still no evidence presenting that Anxa2 is a component for NF-kappa B –DNA/RNA complex. We believe Anxa2 binds to p50 with its N-terminus, and the conformation of the N-terminus is controlled by couples of residue around Lys301. Our previous molecular docking present that the most appropriate binding site of (20S)G-Rh2 shares the same path of Anxa2 N-terminus on the surface of Anxa2, and the conformation of the N-terminus is altered when (20S)G-Rh2 interacts with Anxa2, which inhibits Anxa2-p50 interaction. K301A mutant fails to bind to (20S)G-Rh2 and protects the conformation of the N-terminus as well as its binding capability to p50. So we do not take DNA/RNA binding capability into this mechanism.
